# The ketogenic diet alleviates autoimmune thyroiditis caused by Th17/Treg imbalance by inhibiting the HMGB1/NLRP3 signaling pathway

Yue Luo[1], Hao Gao[1], Mengzhen Wang[1], Zhimin Wang[2], Zhe Jin[3], Nan Song[4]*, Ziyu Liu[5]*, Xiao Yang[5]*

1 School of Graduate Studies at Liaoning University of Traditional Chinese Medicine, Shenyang, China, 2 Department of the First Clinical College, Liaoning University of Traditional Chinese Medicine, Shenyang, China, 3 Department of the Second Clinical College, Liaoning University of Traditional Chinese Medicine, Shenyang, China, 4 Department inflam of Medical Laboratory, Liaoning University of Traditional Chinese Medicine, Shenyang, Liaoning, China, 5 Department of Liaoning Institute of Traditional Chinese Medicine, Liaoning University of Traditional Chinese Medicine, Shenyang, China

☯ These authors contributed equally to this work.
* cpcool@126.com (NS), 894458722@qq.com (ZL), dfziran2023@163.com (XY)

## Abstract

### Background

Autoimmune thyroiditis (AIT), caused by immune-mediated thyroid dysfunction, lacks effective dietary interventions. This study investigates the therapeutic potential of a ketogenic diet (KD) in a murine model of iodine-induced AIT.

### Methods

Sixty 8-week-old NOD.H-$2^{h4}$ mice were randomized into three groups: wild-type mice on a normal diet (WT＋ND), AIT model mice on a normal diet (AIT＋ND), and AIT model mice on a KD (AIT＋KD). AIT was induced in model groups via 0.05% sodium iodide administration for 8 weeks, followed by 4 weeks of KD intervention in the AIT＋KD group. Outcomes included histopathological thyroid changes, serum levels of TgAb/TPOAb, proinflammatory cytokines (MCP-1, TNF-α, IL-1β, IL-6, IL-18, IFN-γ), oxidative stress markers (MDA, SOD, T-AOC), gene expression of NLRP3 inflammasome components (NLRP3, ASC, Caspase-1), Th17/Treg-related factors (RORγt, IL-17, FoxP3, IL-10), and immune mediators (TLR2/4, NF-κB, HMGB1).

### Results

Compared to the WT＋ND group, the AIT＋ND group exhibited increased TgAb/TPOAb levels, enhanced thyroid lymphocytic infiltration, elevated proinflammatory cytokines, and aggravated oxidative stress. Concurrently, this group showed upregulated NLRP3 inflammasome activity, promoted Th17 polarization, activated HMGB1/TLR2/4/NF-κB signaling, and a higher Th17/Treg ratio. Ketogenic diet (KD)

**Data availability statement:** All relevant data are within the manuscript and its Supporting Information files.

**Funding:** This work was supported by the General Program of the National Natural Science Foundation of China (82274455, received by XY); the Youth Program of the National Natural Science Foundation of China (82104805, received by ZW); the Key Research Project of the Liaoning Provincial Department of Education (JYTZD2023196 received by XY, JYTMS20231828 received by GW); the Joint Fund of the Liaoning Provincial Science and Technology Department (2023-MSLH-154 received by ZJ, 2023-MSLH-147 received by YZ); the Yin Yuanping National Famous Traditional Chinese Medicine Expert Inheritance Studio (202205 received by XY); and the Reserve Project of Basic Scientific Research in Liaoning Colleges and Universities (2024-YJTCB-045 received by XY).

**Competing interests:** The authors have declared that no competing interests exist.

intervention reversed all these alterations, effectively suppressing inflammation, oxidative stress, and pathogenic immune pathways.

## Conclusion

KD ameliorates iodine-induced AIT in mice by modulating HMGB1/NLRP3-mediated inflammation, restoring immune balance, and reducing thyroid autoimmunity. These findings support KD as a potential adjuvant therapy for AIT, warranting clinical evaluation.

## Introduction

Autoimmune thyroiditis (AIT) is an organ-specific autoimmune disease that occurs in the thyroid gland. The incidence of AIT is about 7.5%, and the prevalence in women is 8–10 times higher than in men [1,2]. Multiple systematic reviews and meta-analyses have confirmed a direct causal link between hypothyroidism and Hashimoto#39;s thyroiditis and osteoporosis, as well as a significant increase in the risk of thyroid cancer and hearing loss [3–5]. The main clinical manifestations of AIT include inflammatory cell infiltration of the thyroid parenchyma, leading to dense accumulation of lymphocytes, plasma cells, and macrophages, forming germinal centers, and accompanied by thyroid enlargement [6]. There is also an increase in the levels of serum thyroglobulin antibody (TGAb) and thyroid peroxidase antibody (TPOAb) [7]. In recent years, the incidence of AIT has been increasing year by year, and its etiology is related to genetic susceptibility, environment, diet, infection, and other factors [8]. Various causes lead to the destruction of immune tolerance, enabling TPOAb antibody-mediated autoimmune reactions, enhancing lymphocyte infiltration, and leading to the destruction of thyroid cells [9].

The NOD.H-2$^{h4}$ mouse is currently a more ideal spontaneous autoimmune thyroiditis (SAT) animal model. Under high-iodine feeding conditions, these mice show thyroid tissue pathology similar to AIT and exhibit increased levels of TGAb [10]. The NLRP3/Caspase-1 axis, as a key inflammatory signaling pathway in the progression of AIT, promotes the release of inflammatory factors and is involved in the occurrence and development of AIT [11]. NLRP3, as a cytosolic pattern recognition receptor, can form an inflammasome with ASC, activating Caspase-1 and playing an important role in AIT [12]. High mobility group protein Box 1 (HMGB1), as an intracellular binding protein, can emit danger or damage-associated molecular pattern signals and can activate the NLRP3 inflammasome and its downstream inflammatory cascade, playing a key role in inflammation [13]. Studies have found that the concentration of HMGB1 in the serum of AIT patients is positively correlated with the titers of TGAb and TPOAb [14]. Therefore, effectively inhibiting the inflammatory signaling axis of HMGB1/NLRP3/Caspase-1 may be an effective way to treat AIT, and it is speculated to play a role in AIT.

The ketogenic diet (KD) is a dietary structure that is high in fat, low in carbohydrates, and has an appropriate amount of protein. It mainly simulates the state of

fasting, using fat as the main source of energy for the body, decomposing fatty acids in the liver, inducing ketone body production, and replacing glucose oxidation for energy supply [15]. The ketogenic diet was initially proposed as a treatment for epilepsy in the 1920s [16]. In recent years, studies have found that KD has significant effects on obesity, diabetes, and even tumors. Research has found that KD can be used for the adjunctive treatment of AIT. A clinical study compared the effects of low-carbohydrate diets with Normal Diet s on the intervention of Hashimoto's thyroiditis and found that the thyroiditis inflammation in 108 patients who underwent low-carbohydrate diet intervention improved, and the titers of TgAb and TPOAb significantly decreased [17]. The endogenous ketones produced by KD metabolism, especially β-hydroxybutyrate, can exert anti-inflammatory effects by blocking NLRP3-mediated inflammatory reactions [18]. Yao, Y [19] pointed out that KD can reduce the production of pro-inflammatory factors such as TNF-α, IL-6, and IFN-γ by inhibiting the NF-κB signaling pathway, thereby reducing inflammation. Th2 cells secrete cytokines like IL-6 to primarily mediate humoral immunity, promoting B lymphocyte proliferation and, thereby influencing thyroid function [20]. The study showed that TNF-α and IL-6 levels are both increased in HT patients, and they are positively correlated [21]. IFN-γ alone or with other inflammatory cytokines induces MHC I/II on APCs & other cells. It then upregulates adhesion molecules, chemokines & receptors to recruit T cells to inflammation sites [22].

However, whether it can reduce the inflammatory cascade reaction by improving the HMGB1/NLRP3/Caspase-1 signaling axis to play an anti-AIT role still needs further research. Therefore, based on previous research and reports, the purpose of this study was to observe the anti-AIT effect of KD on iodine-induced NOD.H-2$^{h4}$ mice and for the first time discussed its mechanism of action based on the HMGB1/NLRP3/Caspase-1 signaling pathway, providing a theoretical basis for the mechanism study of KD in the treatment of AIT.

## Materials and methods

### Animal

The NOD.H-2$^{h4}$ mice (strain number No. 004447) obtained from Jackson Laboratory, USA, were meticulously bred and housed in the SPF-grade animal experimental center of Liaoning University of Traditional Chinese Medicine. The breeding and housing procedures strictly adhered to approved standards, with the facility being licensed under SYXK (Liao) 2019−0004. The animal study was reviewed and approved by the Institutional Animal Care and Use Committee at Liaoning University of Traditional Chinese Medicine. The experimental protocol received approval from the Ethics Committee of Liaoning University of Traditional Chinese Medicine, bearing approval number 21000042023062.

### Reagents and drugs

Mouse Monocyte Chemotactic Protein 1 ELISA Kit (Catalog Number: JL20304), Mouse Tumor Necrosis Factor Alpha ELISA Kit (Catalog Number: JL10484), Mouse Interleukin 1 Beta ELISA Kit (Catalog Number: JL18442), Mouse Interleukin 6 ELISA Kit (Catalog Number: JL20268), Mouse Interleukin 18 ELISA Kit (Catalog Number: JL20253), Mouse interferon ELISA Kit (Catalog Number: JL51526) were acquired from Shanghai Jianglai Biotechnology Co., Ltd. Lipid Peroxidation MDA Assay Kit (Catalog Number: S0131S), Total Superoxide Dismutase Assay Kit with WST-8 (Catalog Number: S0101S) were acquired from Beyotime Biotechnology. Total antioxidant capacity assay kit (Catalog Number: A015-1–2) were acquired from Nanjing Jiancheng Bioengineering Institue. Rabbit polyclonal antibody to HMGB1 (catalog number: AF7020), Rabbit polyclonal antibody to NLRP3 (catalog number: DF7438), Rabbit polyclonal antibody to ASC (catalog number: DF6304), Rabbit polyclonal antibody to Caspase 1 (catalog number: AF5418), Rabbit polyclonal antibody to TLR2 (catalog number: DF7002), Rabbit polyclonal antibody to TLR4 (catalog number: AF7017), Rabbit polyclonal antibody to NF-kB p65 (catalog number AF5006), Rabbit polyclonal antibody to IL17A (catalog number DF6127), Rabbit polyclonal antibody to Foxp3 (catalog number AF6544), Rabbit polyclonal antibody to IL-10 (catalog number DF6894), Rabbit polyclonal antibody to GAPDH (catalog number AF7021), all the aforementioned reagents were purchased from

Affinity Biosciences Ltd. The ketogenic diet (catalog number XTKD01) was purchased from Jiangsu Xietong Pharmaceutical Biotechnology Co., Ltd. The feed formulations are listed in Table 1,2.

## Model establishment and intervention

NOD.H-2$^{h4}$ mice were selected as the ideal animal model for AIT [10]. A total of 60 8-week-old NOD.H-2$^{h4}$ mice, weighing 20–25 g and equally distributed by sex, were randomly assigned to three experimental groups: WT＋ND group (wild type mice＋normal diet), the AIT＋ND group (model mice＋normal diet, and the AIT＋KD group (model mice＋ketogenic diet),

**Table 1. The feed formulations of normal diet.**

| Normal Diet (ND) | |
| --- | --- |
| **Feed component** | **The content per unit packaging (g)** |
| Crude protein | 50 |
| Crude fat | 27 |
| Coarse fiber | 23.55 |
| Crude ash content | 35 |
| water content | 50 |
| Calcium | 70 |
| Total phosphorus | 50 |
| Lysine | 11.2 |
| Methionine | 4.5 |
| Cysteine | 6.4 |
| Nitrogen free extract | 328 |
| FD&C Red Dye #40 | 0.025 |
| FD&C Yellow Dye #5 | 0.025 |
| FD&C Blue Dye #1 | 0 |
| Total | 610.5 |

**Table 2. The feed formulations of ketogenic diet.**

| Ketogenic Diet (KD) | |
| --- | --- |
| **Feed component** | **The content per unit packaging (g)** |
| Casein, 80 mesh | 100.00 |
| L-Cystine | 1.50 |
| Corn Starch | 0.00 |
| Maltodextrin 10 | 0.00 |
| Sucrose | 0.00 |
| Cellulose | 50.00 |
| Soybean Oil | 25.00 |
| Cocoa Butter | 381.00 |
| Mineral Mix S10026B | 50.00 |
| Vitamin mixV10001C | 1.00 |
| Choline Bitartrate | 2.00 |
| FD&C Red Dye #40 | 0.025 |
| FD&C Yellow Dye #5 | 0.025 |
| FD&C Blue Dye #1 | 0 |
| Total | 610.5 |

with 20 mice in each group. The WT + ND and ND groups were fed a normal diet, while the AIT + KD group was fed a keto-genic high-fat diet. The mice were provided with free access to 0.05% sodium iodide with double-distilled water and were developed into an AIT animal model after 8 weeks [23,24]. Interrupted after 8 weeks of manufacturing AIT model for NaI. Tissue samples were collected after 4 weeks of intervention.

## Sample collection

All experimental procedures adhered to institutional ethical guidelines and the 3Rs principles. Mice were deeply anes-thetized via intraperitoneal sodium pentobarbital (65 mg/kg; 10 mL/kg volume [25]), which resulted in their death. Blood was collected from the orbital vein using sterilized capillaries post-mydriasis (0.5% phenylephrine), while the animals were in a state of deep anesthesia to minimize their suffering. Serum was separated by centrifugation (2000 × g, 15 min) after coagulation and stored in 1.5 mL tubes at −80°C. Thyroid glands were isolated through midline cervical dissection: laryngeal-tracheal complexes were exposed following blunt submandibular gland retraction, with meticulous preservation of parathyroids. Tissues were either fixed in 4% paraformaldehyde or snap-frozen in liquid nitrogen; RT-qPCR samples were preserved with 1 mL DNA extraction reagent prior to −80°C storage. Western blot specimens were directly immersed in liquid nitrogen. Environmental enrichment (nesting materials, social housing) and power analysis-driven sample minimi-zation ensured welfare optimization. The study received ethical approval and complied with ARRIVE guidelines.

## HE staining

Mice were fasted but allowed free access to water for 12 hours before sampling. Thyroid tissues were quickly dissected and fixed in 4% paraformaldehyde for 36 hours. After embedding in paraffin, sections were stained with hematoxylin and eosin. The slides were then mounted with neutral gum and used to observe the degree of inflammatory cell infiltration in the thyroid tissues.

## Immunohistochemistry

The thyroid wax was dewaxed with xylene, dewaxed with anhydrous ethanol, then put into citrate buffer for antigen repair, 3% $H_2O_2$ was used to eliminate the activity of endogenous peroxidase, incubated at 37 °C for 1 h, diluted antibody was added and incubated all night at 4 °C. Then, the secondary antibodies were incubated in a wet box at 37 °C for DAB stain-ing, hematoxylin re-staining, 1% hydrochloric acid alcohol differentiation and neutral resin sealing.

## Immunofluorescence Staining

After deparaffinization, thyroid sections were performed with antigen retrieval in 0.01 M citrate buffer solution. Sections were then incubated with a primary antibody at 4 °C overnight, followed by incubation with secondary antibody. A fluores-cence microscope was used to obtain images of the tissues.

## ELISA assay

The mice were sacrificed, and approximately 1.5 mL of blood was collected from the eyeball. The blood was left to stand at room temperature for 2 hours and then centrifuged at 3500 rpm for 15 minutes (8 cm radius). The supernatant was separated. ELISA kits were used to measure the levels of TGAb, TPOAb, MCP-1, TNF-α, IL-1β, IL-6, IL-18, IFN-γ, MDA, SOD, T-AOC in the serum [26].

## RT-qPCR assay

According to the instructions of Trizol reagent, the total RNA of thyroid tissue was extracted, the absorbance of 260 nm RNA was determined by UV spectrophotometer, and the purity and concentration of RNA were calculated. The genomic

DNA reaction was removed by reverse transcription kit, and the reaction system was prepared on ice. After 2 min was incubated at 42 °C, 5 min was incubated at 42 °C for 15 min and incubated at 85 °C to synthesize cDNA. Using β-actin as the internal reference mRNA, SYBR Green was used according to the pre-denaturation at 95 °C, annealing at 60 °C for 40 cycles, reaction at 95 °C at 15 °C, min, 95 at 60 °C and reaction at 50 °C for 30s. The primer sequences for amplifying NLRP3, ASC, Caspase-1, RORγt, IL-17, FoxP3 and IL-10 were designed and synthesized by Dalian Bao Biological Company (see table for primer sequences). Then, the results were analyzed, and the relative expression levels of the target genes in each group were determined using the $2^{-\Delta\Delta CT}$ method in Table 3.

## Western blot analysis

Thyroid samples were homogenized using a glass homogenizer. The protein lysate was added and kept on ice for 30 minutes, followed by centrifugation at 12,000 rpm for 5 minutes to collect the supernatant. Protein quantification was performed using a UV spectrophotometer by colorimetric assay, and a standard curve was plotted to calculate the corresponding protein concentration. SDS-PAGE gel was prepared, and the samples were denatured by boiling in the loading buffer for 5 minutes. SDS-PAGE electrophoresis and transfer to a membrane were performed. After transfer, the membrane was washed with TBST once. Blocking was carried out with 5% skim milk for 1 hour, followed by washing with TBST and incubation with primary antibodies at 4 °C. The next day, after washing the membrane with TBST three times, it was incubated with secondary antibodies at room temperature on a shaking platform for 1 hour. After three washes with TBST, the membrane was subjected to enzymatic visualization using a chromogenic substrate. The image J analysis software was used for analysis.

## Flow cytometry

$1 \times 10^6$ cells were resuspended in $1 \times$ binding buffer solution. For surface staining, cells were labeled with anti-CD4-FITC antibody for 30 minutes at 4 °C in the dark. Subsequently, cells were fixed with 4% paraformaldehyde and permeabilized using a cell permeabilization Kit for a duration of 35 min. Thereafter, fora duration of 30 minutes at 4 °C in the dark.

**Table 3. Primer Sequences.**

| Genes | Primer name | Primer sequence (5' ->3') | Product length / bp |
|---|---|---|---|
| NLRP3 | NLRP3-F | ATTACCCGCCCGAGAAAGG | 141 |
| | NLRP3-R | TCGCAGCAAAGATCCACACAG | |
| ASC | ASC-F | CTTGTCAGGGGATGAACTCAAAA | 154 |
| | ASC-R | GCCATACGACTCCAGATAGTAGC | |
| Caspase-1 | Caspase-1-F | ACAAGGCACGGGACCTATG | 237 |
| | Caspase-1-R | TCCCAGTCAGTCCTGGAAATG | |
| RORγt | RORγt - F | GACCCACACCTCACAAATTGA | 137 |
| | RORγt - R | AGTAGGCCACATTACACTGCT | |
| IL-17 | IL-17 – F | TTTAACTCCCTTGGCGCAAAA | 165 |
| | IL-17 – R | CTTTCCCTCCGCATTGACAC | |
| FoxP3 | FoxP3 - F | CCCATCCCCAGGAGTCTTG | 183 |
| | FoxP3 - R | ACCATGACTAGGGGCACTGTA | |
| IL-10 | IL-10 – F | GCTCTTACTGACTGGCATGAG | 105 |
| | IL-10 – R | CGCAGCTCTAGGAGCATGTG | |
| β-actin | β-actin-F | GGCTGTATTCCCCTCCATCG | 154 |
| | β-actin-R | CCAGTTGGTAACAATGCCATGT | |

Following this, cells were washed twice with PBS and acquired using an LSR Fortessa four-laser flow cytometer. The data was then analyzed using FlowJo software.

## Statistical analysis

All experimental data in this study were analyzed using Excel, SPSS 22.0, and GraphPad Prism 8 statistical software. Initially, the normality of data distribution was assessed via the Shapiro-Wilk test. For normally distributed data, independent samples t-tests were employed for intergroup comparisons; for non-normally distributed data, nonparametric Mann-Whitney U tests (rank-sum tests) were utilized. In comparisons involving multiple groups, one-way analysis of variance (ANOVA) was applied when data conformed to normal distribution with homogeneous variances, while the nonparametric Kruskal-Wallis H test (rank-sum test) was used when variances were heterogeneous.

Statistical significance was defined as $P < 0.05$ (two-tailed tests). All analytical processes were based on raw data, with graphical and numerical outputs generated through SPSS 22.0. Final statistical conclusions were drawn based on significance levels ($P < 0.05$) to determine whether differences between experimental and control groups were statistically meaningful. Quantitative results are presented as mean ± standard deviation (Mean ± SD), with intergroup comparisons visually supplemented by figures/tables as required.

## Results

### Ketogenic diet alleviated autoimmune thyroiditis induced by high iodine intake in mice

Fig 1A shows the schematic design of experimental autoimmune thyroiditis. Compared with the WT + ND group, serum TgAb and TPOAb levels were increased, and thyroid volume was increased ($P < 0.01$) in the AIT + ND (Fig 1B, 1C and supplementary table 1). After KD treatment, serum TgAb and TPOAb levels decreased, and thyroid volume decreased significantly ($P < 0.05$, $P < 0.01$). There were few lymphocytes and plasma cells infiltrated among the follicles. In contrast, in the AIT + ND group, many thyroid follicles were destroyed, atrophic and heavily infiltrated with lymphocytes, and follicles were reduced after KD treatment. KD significantly reduced the rating and score of inflammation ($P < 0.05$, $P < 0.01$) in AIT + ND group (Fig 1D, 1E). In addition, the body weight of the mice was continuously monitored during the diet intervention, and the results showed that the body weight of the AIT + KD group was significantly lower than that of the AIT + ND group (Fig 1F). The results showed that KD alleviated thyroiditis and goiter in AIT + ND group.

### Effects of ketogenic diet on serum cytokines and oxidative stress markers

The experimental results showed (Fig 2A–2F, S2 File) that the serum levels of MCP-1, TNF-α, IL-1β, IL-6, IL-18 and IFN-γ were significantly increased in AIT + ND group($P < 0.01$). However, all inflammatory factors were decreased ($P < 0.05$, $P < 0.01$) after KD intervention. We also examined the expression levels of oxidative stress markers MDA, SOD, and T-AOC and found that SOD and T-AOC levels decreased and MDA levels increased ($P < 0.01$) in the AIT + ND group, and the trend was reversed after KD treatment (Fig 2G– 2I, S3 File). Elevated serum concentrations of MCP-1, TNF-α, IL-1β, IL-6, IL-18, IFN-γ, and MDA indicate acute inflammatory responses, whereas reduced levels of SOD and T-AOC suggest a correlation between impaired antioxidant capacity and exacerbated inflammatory processes. Collectively, these findings imply that KD intervention attenuates inflammatory responses and mitigates oxidative stress in the AIT + ND group. High values of MCP-1, TNF – α, IL-1 β, IL-6, IL-18, IFN – γ, and MDA indicate acute inflammatory response, while low values of SOD and T-AOC indicate a close correlation between decreased antioxidant capacity and intensified inflammatory response. In summary, KD can alleviate the inflammatory response and improve oxidative stress in the ND group.

### KD Decreased the Expression of HMGB1, TLR2, TLR4, NF-κB

The protein expression levels of HMGB1, TLR2, TLR4 and NF-κB in WT + ND group, AIT + ND group and AIT + KD group were detected by Western blot, respectively. The results of Western blot analysis (Fig 3A) showed that the protein

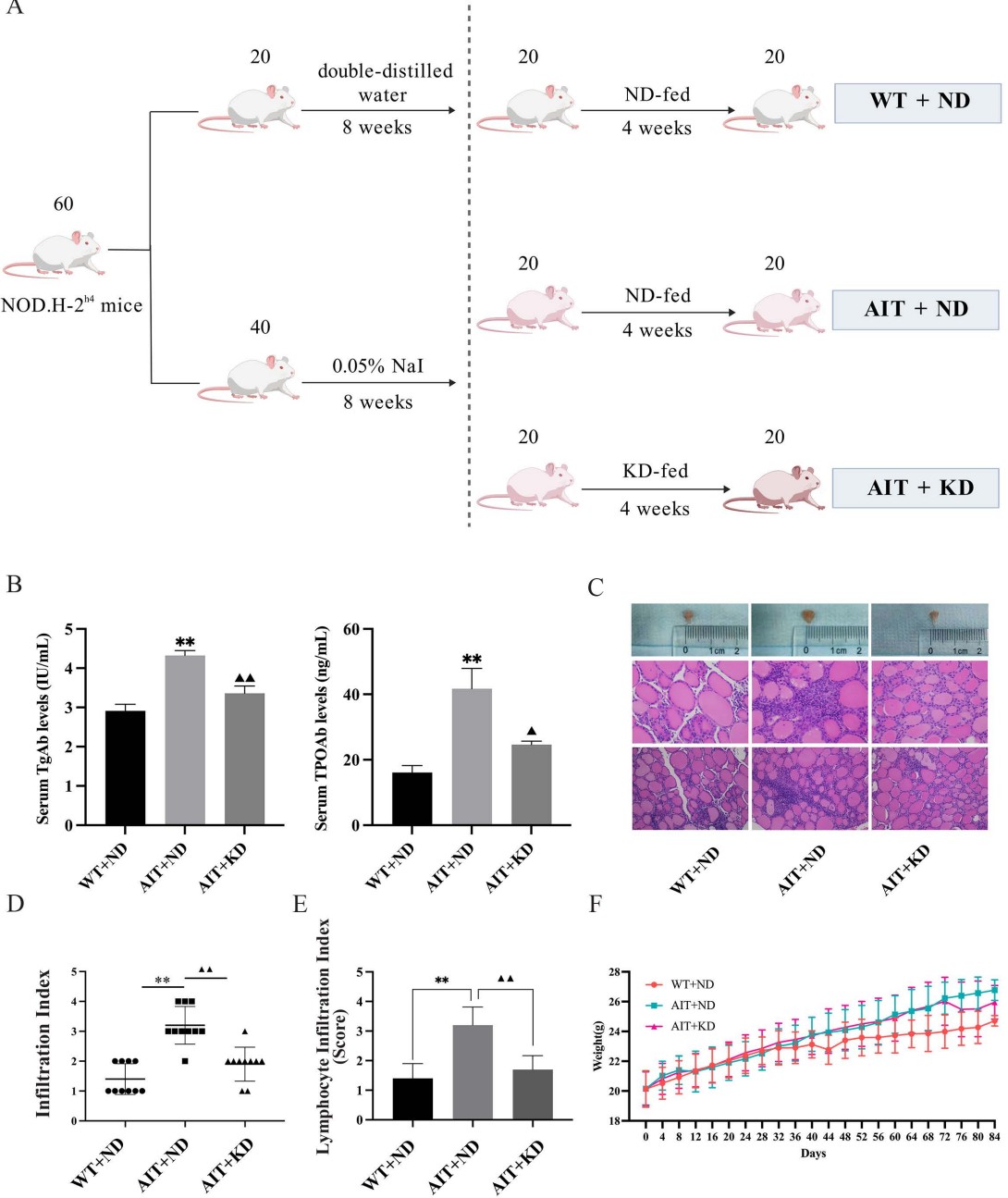

**Fig 1. Ketogenic diet attenuates autoimmune thyroiditis induced by high iodine intake in mice. (A)** Schematic design of experimental autoimmune thyroiditis (n = 10). **(B)** Serum TgAb and TPOAb levels (n = 10). **(C)** Representative histological image of the thyroid of a mouse with autoimmune thyroiditis (n = 10). **(D)** lymphocyte infiltration rating and **(E)** score (n = 20). **(F)** Body weight of mice was continuously monitored during the dietary intervention (n = 20). P-value from one-way ANOVA test. The data are expressed as the mean ± SD. **$P < 0.01$ versus WT + ND group; ▲▲$P < 0.01$ versus AIT + ND group.

expression of HMGB1, TLR2, TLR4, and NF-κB ($P < 0.01$) in the AIT + ND group was significantly higher than that in the WT + ND group, and decreased after treatment with KD compared with the AIT + ND group ($P < 0.01$). The IHC results (Fig 3B) were consistent with these results.

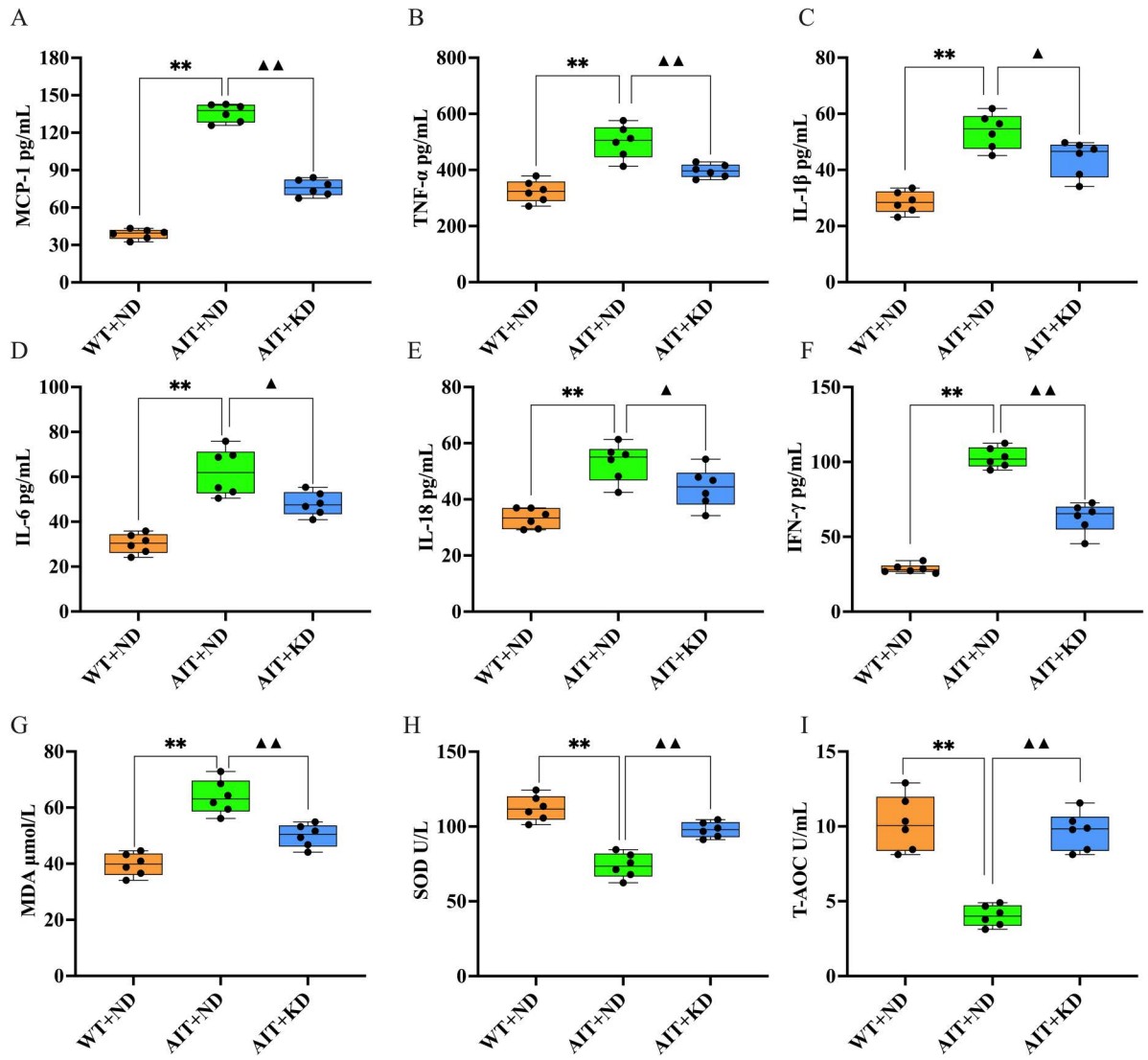

**Fig 2. Effect of ketogenic diet on serum cytokines and oxidative stress markers in ND group.** The secretion levels of **(A)** MCP-1, **(B)** TNF-α, **(C)**IL-1β, **(D)** IL-6, **(E)** IL-18, **(F)** IFN-γ, **(G)** MDA, **(H)** SOD and **(I)** T-AOC were measured by ELISA. The data are expressed as the mean±SD (n=6). P-value from one-way ANOVA test. $**P < 0.01$ versus WT+ND group; $\blacktriangle\blacktriangle P < 0.01$ versus AIT+ND group.

## KD Decreased the Expression of NLRP3/ASC/Caspase-1 Signaling Pathway

The next step is to elucidate the underlying mechanism of thyroiditis in KD-mediated remission in AIT+ND group. As shown in Fig 4A, 4B, mRNA and protein expression levels of NLRP3, ASC, and Caspase-1 in the AIT+ND group was significantly higher than that in the WT+ND group ($P < 0.05$, $P < 0.01$), and AIT+KD effectively reduced the expression of NLRP3, ASC, and Caspase-1 mRNA and protein ($P < 0.01$). The expressions of ASC and Caspase-1 were further detected by immunofluorescence (Fig 4C), and it was found that the content of ASC and Caspase-1 in the AIT+ND group was higher than that in the WT+ND group, and the content of ASC and Caspase-1 was significantly decreased after KD treatment. These results indicate that KD can reduce the expression of NLRP3/ASC/Caspase-1 signaling pathway, thereby alleviating AIT.

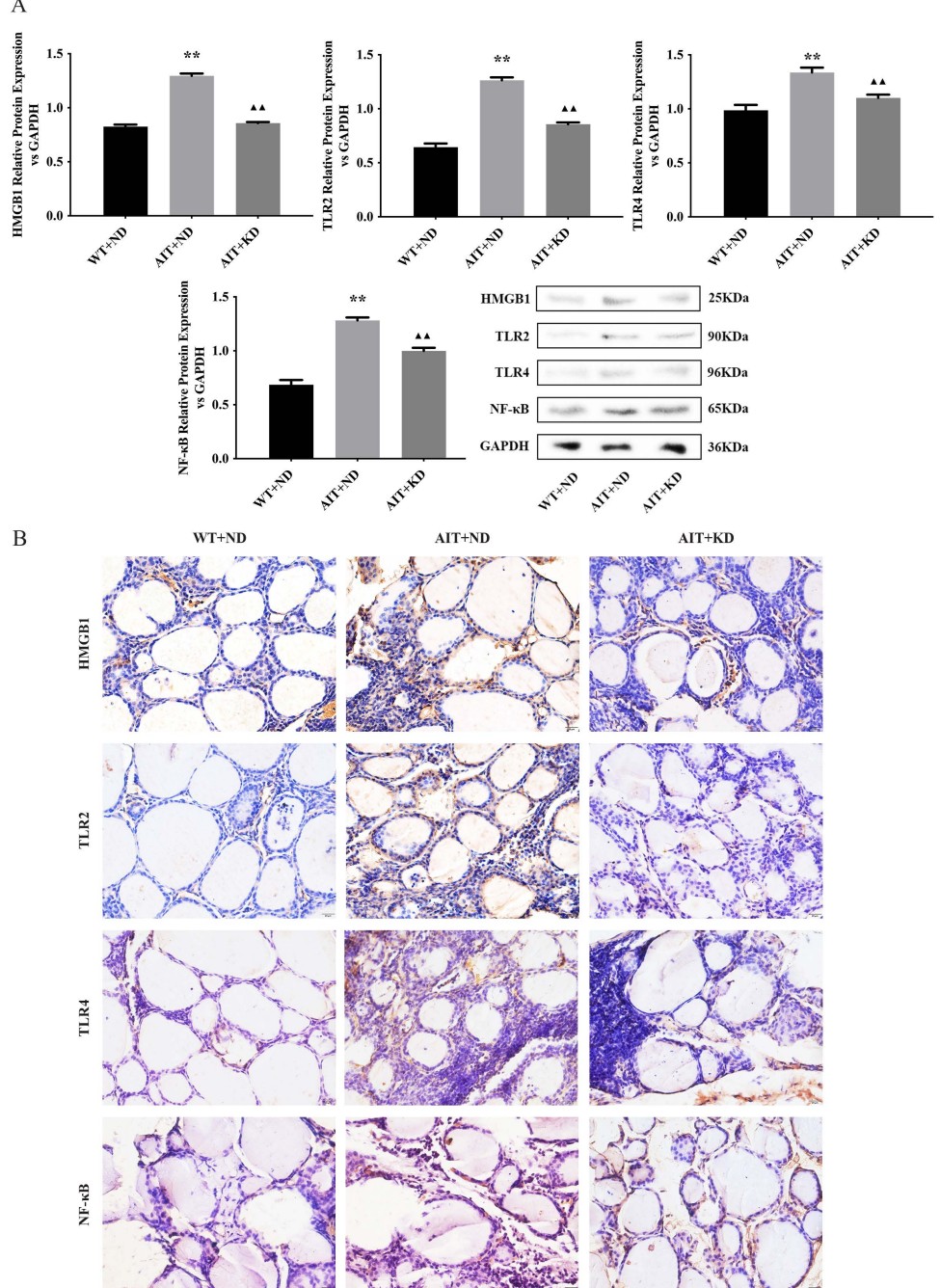

**Fig 3. KD Decreased the Expression of HMGB1, TLR2, TLR4, NF-κB. (A)** Expression levels of protein in thyroid tissues of mice in each group. **(B)** Expression of IHC staining in thyroid tissues of mice in each group. The data are expressed as the mean ± SD (n = 3). P-value from one-way ANOVA test. **$P < 0.01$ versus WT + ND group; ▲▲$P < 0.01$ versus AIT + ND group.

## KD Rescued Th17/Treg Imbalance in mice

We used flow cytometry to determine percentage Th-cell subsets. The changes of Th17 and Treg cell subsets in AIT + ND group were significantly higher than those in WT + ND group, and the proportion of Th17 and Treg cell subsets decreased

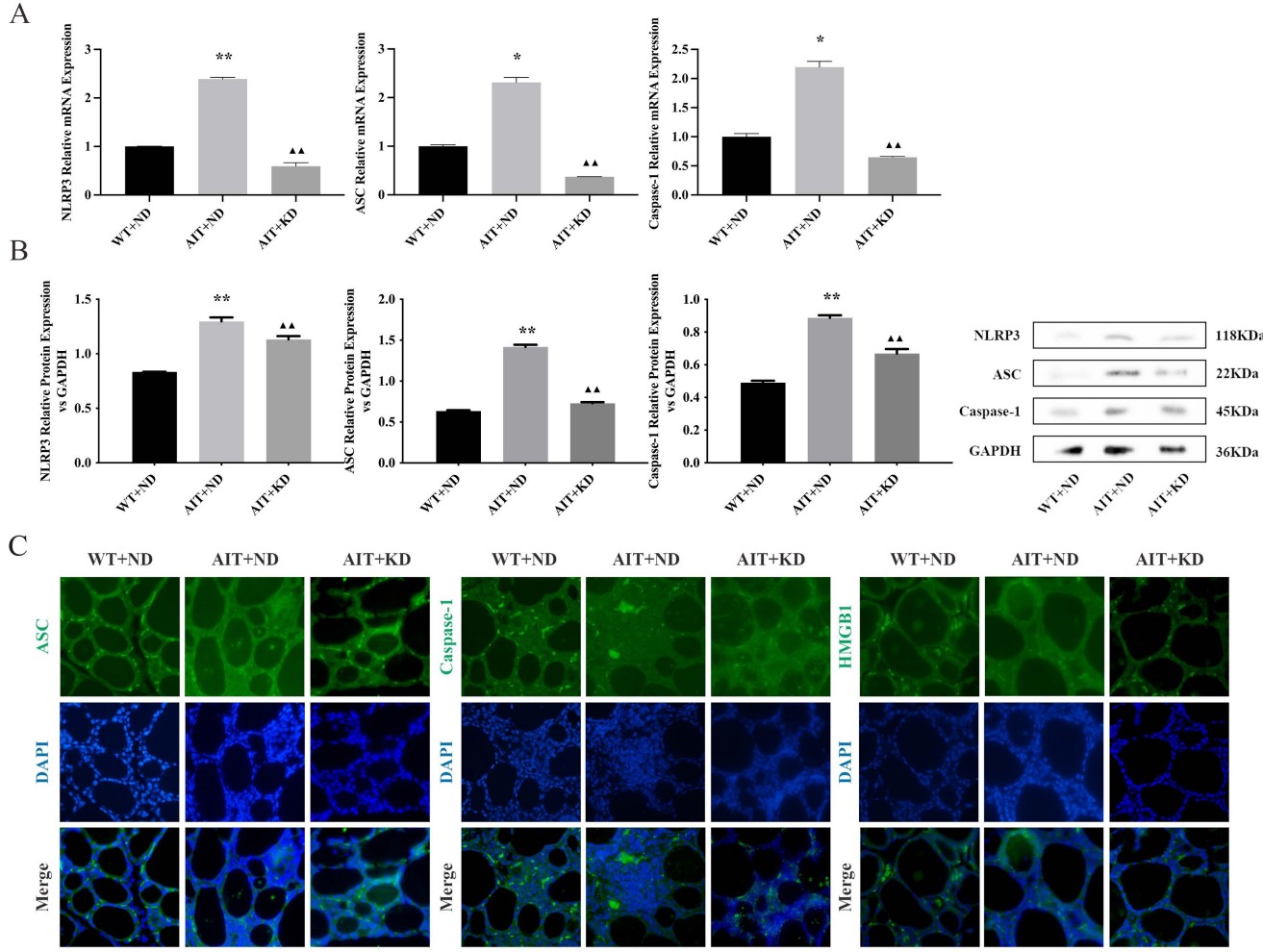

**Fig 4. KD Decreased the Expression of NLRP3/ASC/Caspase-1 Signaling Pathway. (A)** Expression levels of NLRP3, ASC and Caspase-1 mRNA in thyroid tissues of mice in each group. **(B)** Expression levels of NLRP3, ASC and Caspase-1 protein in thyroid tissues of mice in each group. **(C)** Expression levels of ASC, Caspase-1 and HMGB1 Immunofluorescence staining in thyroid tissues of mice in each group. Differences among the groups were analysed using ANOVA. The data are expressed as the mean ± SD (n = 3). **$P < 0.01$ versus WT + ND group; ▲▲$P < 0.01$ versus AIT + ND group.

significantly after KD treatment (Supplementary figure 1). In addition, changes in the levels of Th17, Treg-specific transcription factors RORγt, IL-17, FoxP3, and IL-10 were determined by RT-qPCR and Western blot, respectively (Fig 5A, 5B). The mRNA and protein expressions of RORγt and IL-17 were significantly increased in the AIT + ND group, which were significantly reduced by KD ($P < 0.05$, $P < 0.01$). The mRNA and protein expressions of FoxP3 and IL-10 in WT + ND group were significantly lower than those in AIT + ND group, and increased after KD treatment ($P < 0.05$, $P < 0.01$).

## Discussion

Autoimmune thyroiditis (AIT) is a condition that leads to the accumulation of inflammatory cells such as T cells, B cells, macrophages, and plasma cells around the thyroid, causing immune system dysfunction. This study refers to international modeling standards and uses excessive iodine to induce the NOD.H-2$^{h4}$ model [10]. TPOAb and TgAb are characteristic autoantibodies of AIT, and their expression levels are commonly used as important indicators for the

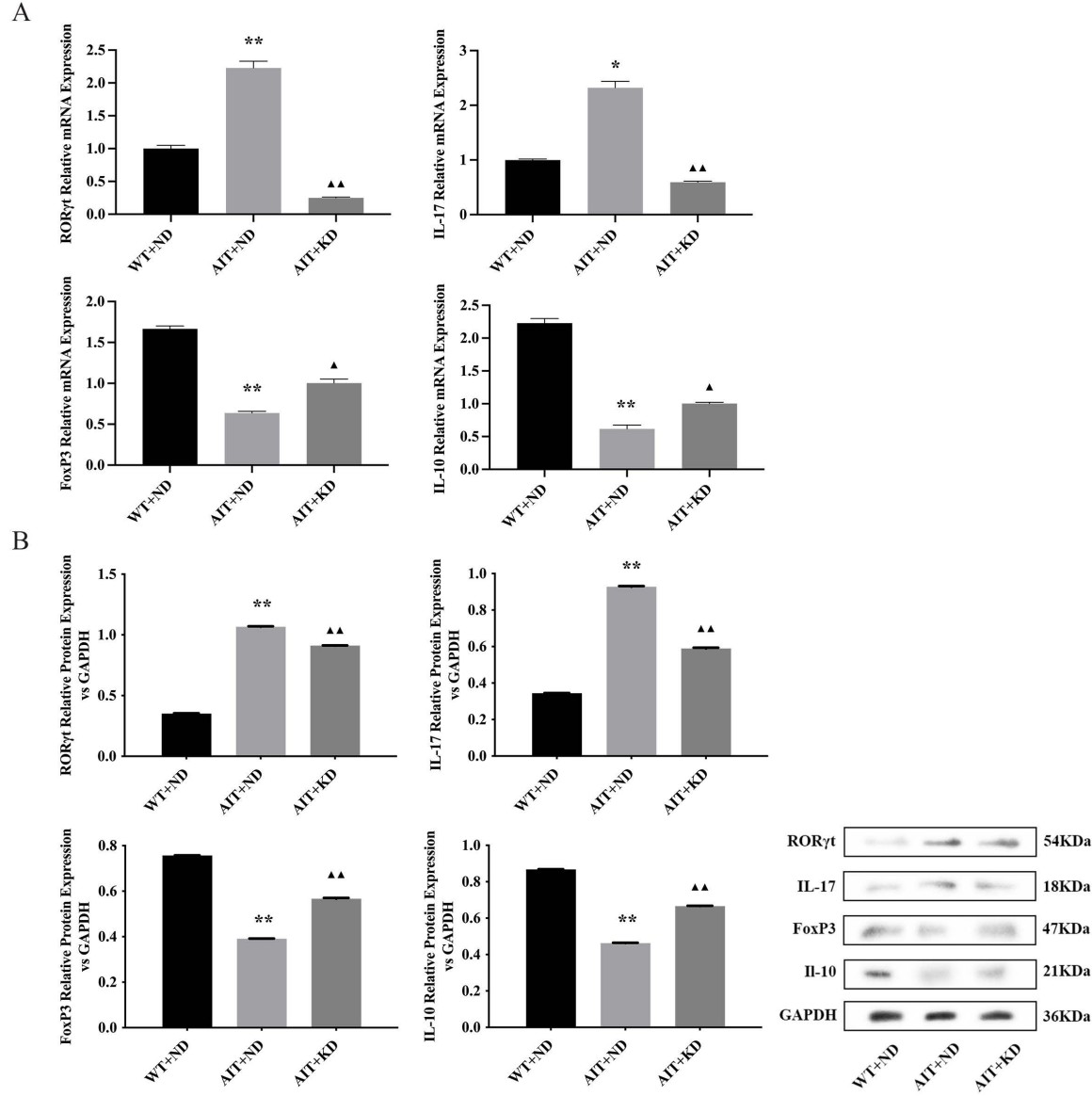

**Fig 5. KD Rescued Th17/Treg Imbalance in mice. (A)** Expression levels of RORγt, IL-17, FoxP3, IL-10 mRNA in thyroid tissues of mice in each group. **(B)** Expression levels of RORγt, IL-17, FoxP3, IL-10 protein in thyroid tissues of mice in each group. P-value from one-way ANOVA test. The data are expressed as the mean±SD (n=3). **$P<0.01$ versus WT+ND group; ▲▲$P<0.01$ versus AIT+ND group.

diagnosis of AIT in clinical practice [27]. This study showed that the levels of TPOAb and TgAb in the serum of mice in the Model group were increased, and the thyroid follicular epithelial cells were disordered and lost their integrity, accompanied by widespread lymphocyte infiltration, indicating that the AIT model mice were successfully constructed. Compared with the Model group, the integrity of thyroid follicular epithelial cells in the thyroid tissue of mice in the AIT+KD group was restored, lymphocyte infiltration was reduced, inflammation was alleviated, pathological changes in thyroid tissue were reduced, and the levels of TPOAb and TgAb in serum were significantly lowered, indicating that KD could improve thyroid damage in AIT mice.

Inflammation is closely related to the occurrence and development of Hashimoto's thyroiditis. The infiltration of inflammatory cells and the increased release of pro-inflammatory factors can lead to the destruction and apoptosis of thyroid cells. Studies have shown that KD can inhibit the release of pro-inflammatory factors and reduce the body's inflammatory response [28]. In the present study, the efficacy of the KD intervention was validated by defining the KD formulation in detail. Outcome variables were assessed using multi-dimensional indices, including both serum and tissue samples, as well as protein and mRNA levels, thereby ensuring the reliability of the results. This study found that the levels of inflammatory markers such as MCP-1, TNF-α, IFN-γ, and IL-6 in the serum of AIT mice were significantly increased, indicating a significant inflammatory response, and KD could reduce the inflammatory levels in AIT mice, indicating that KD could participate in the protective effect of AIT by regulating inflammatory factors. MDA is a lipid peroxide, and excessive accumulation of MDA in the body is one of the main reasons for energy supply damage. Studies have found that the ketogenic diet can reduce the level of MDA [29]. In this study, the serum MDA in the AIT + KD group was significantly lower than that in the Model group, indicating that KD reduced serum peroxidation. Oxidative stress is usually defined as a state of excessive presentation of reactive oxygen species (ROS) due to overproduction and/or impaired elimination, and as one of the pathogenic mechanisms of AIT, when thyroid tissue synthesizes thyroid hormones, ROS are produced, damaging thyroid cells. The KD pattern mainly meets the body's metabolic energy needs by promoting the oxidation of fatty acids in liver cells to produce ketones, with ketones such as β-hydroxybutyrate (β-hydroxybutyrate, β-OHB) and acetoacetate being the main components, transported through the blood to tissues such as the heart, brain, kidneys, and skeletal muscles [30]. The serum SOD and T-AOC contents in the Model group were low, indicating an imbalance of oxidative products and antioxidant substance content in AIT mice, which could be reversed after KD intervention. The enzyme-linked immunosorbent assay (ELISA) is widely used for interleukin detection in biomedical research due to its high throughput, operational convenience, and relatively low cost. However, this method has inherent limitations that may introduce measurement bias, such as ELISA quantifies the total protein level of ILs but cannot differentiate between biologically active cytokines and their inactive precursors, degradation products, or cytokine-antibody complexes, furthermore especially when compared with alternative techniques such as FCM, qRT-PCR, and Western blotting. Thus, the aforementioned techniques were employed to address these limitations.

High mobility group box protein B1 (HMGB1) is a member of the HMG family. It can be actively secreted by necrotic cells or passively released by immune cells into the extracellular matrix, acting as a damage-associated molecular pattern (DAMP) molecule to trigger pro-inflammatory responses [31]. Subsequently, by binding to Toll-like receptors 2/4/9 (TLR2/4/9) or the receptor for advanced glycation end products (RAGE), it plays an important role in inflammatory diseases [32]. HMGB1 triggers immune responses by binding to TLR2 and TLR4, and TLR2 and TLR4 then activate the downstream nuclear factor κB (NF-κB) signaling pathway through myeloid differentiation primary response gene 88 (MyD88), thereby regulating the production of various inflammatory mediators and cytokines [33]. Zhang et al. have shown that HMGB1 can activate NF-κB in human monocytes [34]. HMGB1 is involved in the occurrence and development of AIT. Literature has confirmed that in the iodine-induced NOD.H-2$^{h4}$ mouse model of AIT, the expression of HMGB1 in thyroid tissue and serum is significantly increased, and by mediating the activation of the TLRs/NF-κB pathway, it promotes inflammation and autoimmunity, causing inflammatory infiltration and exacerbating the severity of AIT [35]. Na Liu et al. reported that HMGB1 activates the NLRP3 inflammasome through the TLR4 receptor, thereby promoting caspase-1 activation and IL-1β secretion to accelerate disease progression [36]. In liver diseases, HMGB1 activates NLRP3 inflammasome by binding to exogenous and endogenous ligands, which in turn activates Caspase-1, promotes the maturation of IL-1β and IL-18, and ultimately aggravates liver injury [37]. The results of this study show that compared with the normal group, the protein expression of HMGB1, TLR2, TLR4, and NF-κB in the thyroid tissue of mice in the Model group was increased, while KD could inhibit the expression of HMGB1, TLR2, TLR4, and NF-κB proteins. Alzokaky AA and others have found that HMGB1 can activate the promotion of NLRP3 inflammasome formation, induce downstream Caspase-1, and trigger a cascade of inflammatory reactions [38]. Recent research has found that the NLRP3 inflammasome plays

an important role in AIT inflammation and damage, and the activation of the NLRP3 inflammasome can promote the expression of apoptotic-associated speck-like protein (ASC), as well as the expression and release of Caspase-1 and downstream inflammatory factors IL-1β and IL-18, inducing excessive inflammatory responses and tissue damage in the body, and exacerbating the pathological process of AIT [11]. The results of this study show that compared with the normal group, the NLRP3/ASC/Caspase-1 signaling pathway and the expression of IL-1β and IL-18 mRNA and proteins in the thyroid tissue of mice in the Model group were increased, while KD could improve the inflammatory damage in mice. As a high-fat and low-carbohydrate dietary pattern, KD plays a multi-target anti-inflammatory role by inducing ketosis [39]. Further research has found that the KD can reduce the production of pro-inflammatory cytokines, including tumor necrosis TNF-α, IL-1β, and IL-6, as well as lower the expression of molecular patterns such as TLR4 and NF-κB [40]. This indicates that KD not only inhibits the HMGB1-TLR/NF-κB axis but also achieves the therapeutic goal of AIT by blocking the activation of NLRP3 inflammasome and the release of downstream cytokines.

The Th17/Treg axis is closely related to AIT. FOXP3 is a transcription factor specifically expressed by Treg cells, and Treg cells participate in the suppression of inflammation in various immune diseases by secreting anti-inflammatory cytokines such as IL-10 [41]. Th17 expresses the transcription factor RORγt and mediates inflammatory responses by secreting IL-17 [42]. Under normal circumstances, Th17 and Treg cells are in a relative balance to maintain the stability of the immune system in the body. When this balance is disrupted, autoimmune inflammation occurs. Chaudhry and others have found that FoxP3 inhibits RORγt-mediated IL-17A mRNA transcription by directly binding to RORγt, regulating Th17 cell function and affecting Th17/Treg balance [43,44]. This study further explored the effect of KD on the immune imbalance of Th17/Treg in AIT mice and found that there was an immune imbalance phenomenon, with significant disorders in cell differentiation transcription and effector factors RORγt, FoxP3, IL-17, and IL-10 at the gene and protein levels, consistent with the above reports; after KD intervention, it could inhibit RORγt and IL-17 levels, and upregulate FoxP3 and IL-10 levels, indicating that KD restores Th17/Treg immune balance, regulates the pro-inflammatory/anti-inflammatory axis, and reduces immune inflammation in AIT mice.

KD can significantly improve thyroid pathological damage and inflammatory response in mice with AIT by inhibiting HMGB1/NLRP3/Caspase-1 inflammatory pathway and regulating Th17/Treg immune balance. Although KD has shown significant anti-inflammatory and immunomodulatory effects in animal models, its clinical application potential still needs to be verified by rigorously designed human trials. In addition, current studies are limited to specific mouse strains, and future studies can be extended to multiple mouse strains to capture multiple response patterns that match the diversity of human AIT.

In addition, current studies are limited to specific mouse strains and specificity-related bias in ELISA may mislead the interpretation of inflammatory response intensity in autoimmune thyroiditis (AIT) models, and future studies can be extended to multiple mouse strains to capture multiple response patterns that match the diversity of human AIT.

Using the international iodine-excess NOD.H-2h4 standard, we established a rigorous AIT model and provide the first integrated evidence that a ketogenic diet simultaneously suppresses the HMGB1–TLR/NF-κB–NLRP3 inflammasome axis and rebalances the Th17/Treg compartment, markedly attenuating thyroid damage and circulating autoantibody levels. These findings offer a novel dietary strategy for AIT management. Nevertheless, the present work is confined to in vivo proof-of-concept, and the precise molecular targets and regulatory hierarchies remain to be definitively elucidated in future studies.

## Conclusions

KD, characterized by high fat and low carbohydrate intake, can positively impact AIT through mechanisms such as inhibiting inflammatory responses, reducing autoantibody levels, and modulating immune function. Although the precise mechanisms and efficacy of KD in AIT treatment remain incompletely understood, existing evidence suggests its ability to ameliorate disease symptoms and alleviate inflammatory reactions (Fig 6).

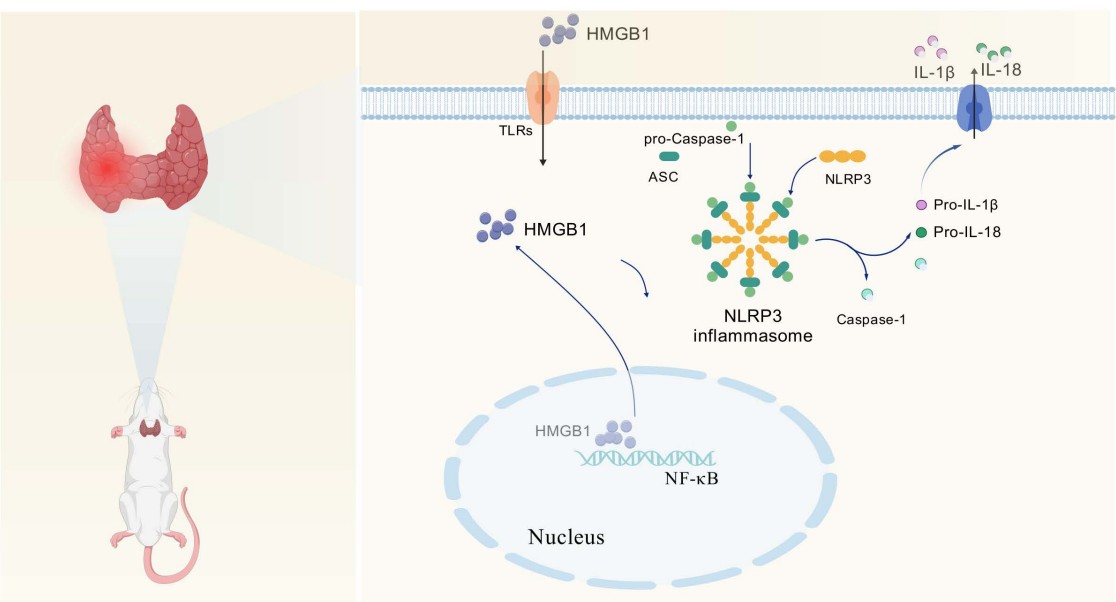

**Fig 6. Roadmap to molecular mechanisms of action.**

Nonetheless, despite some research exploring the effects of the ketogenic diet in AIT treatment, several unresolved issues and directions for further investigation persist. Future studies should delve deeper into the impact of the ketogenic diet on the pathogenesis of AIT, including its mechanisms of action on immune regulation, inflammatory responses, auto-antibody levels, and other relevant aspects. Simultaneously, conducting larger-scale, randomized controlled clinical trials will be crucial to evaluate the efficacy and safety of the ketogenic diet in AIT patients and determine the optimal dietary protocols and timing of implementation. Long-term assessments of patients adopting the ketogenic diet as a therapeutic approach for AIT should be conducted to comprehensively understand its effects on long-term efficacy, metabolic status, cardiovascular health, and other pertinent factors.

## Supporting information

**S1 File. Ketogenic diet attenuates autoimmune thyroiditis induced by high iodine intake in mice.**
(PDF)

**S2 File. Effect of ketogenic diet on serum cytokines and oxidative stress markers in ND group.**
(PDF)

**S3 File. TKD Decreased the Expression of HMGB1, TLR2, TLR4, NF-κB.**
(PDF)

**S4 File. KD Decr eased the Expr ession of NLRP3/ASC/Caspase-1 Signaling Pathway.**
(PDF)

**S5 File. KD Rescued Th17/Tr eg Imbalance in mice.**
(PDF)

**S1 Fig. The percentages of Treg and Th17 cells in the spleen was detected by flow cytometry.** (A) Flow cytometry plots of Th17 cells in the spleens of mice in each group. (B) Flow cytometry plots of FoxP3 cells in the spleens of mice in each group.
(JPG)

## Author contributions

**Conceptualization:** Zhimin Wang.

**Data curation:** Yue Luo, Hao Gao, Mengzhen Wang.

**Formal analysis:** Yue Luo, Hao Gao, Mengzhen Wang.

**Funding acquisition:** Xiao Yang.

**Investigation:** Ziyu Liu.

**Methodology:** Zhe Jin.

**Project administration:** Nan Song, Ziyu Liu.

**Resources:** Xiao Yang.

**Supervision:** Xiao Yang.

**Writing – original draft:** Yue Luo, Hao Gao, Mengzhen Wang.

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
