## [Decision Letter · Decision Letter 0]

16 May 2025

PONE-D-25-04384The Ketogenic Diet Alleviates Autoimmune Thyroiditis Caused by Th17/Treg Imbalance by Inhibiting the HMGB1/NLRP3 Signaling PathwayPLOS ONE

Dear Dr. Gao,

Thank you for submitting your manuscript to PLOS ONE. After careful consideration, we feel that it has merit but does not fully meet PLOS ONE’s publication criteria as it currently stands. Therefore, we invite you to submit a revised version of the manuscript that addresses the points raised during the review process.

Please submit your revised manuscript by Jun 30 2025 11:59PM. If you will need more time than this to complete your revisions, please reply to this message or contact the journal office at plosone@plos.org. . Please include the following items when submitting your revised manuscript:

If applicable, we recommend that you deposit your laboratory protocols in protocols.io to enhance the reproducibility of your results. Protocols.io assigns your protocol its own identifier (DOI) so that it can be cited independently in the future. For instructions see: https://journals.plos.org/plosone/s/submission-guidelines#loc-laboratory-protocols. Additionally, PLOS ONE offers an option for publishing peer-reviewed Lab Protocol articles, which describe protocols hosted on protocols.io. Read more information on sharing protocols at . Additionally, PLOS ONE offers an option for publishing peer-reviewed Lab Protocol articles, which describe protocols hosted on protocols.io. Read more information on sharing protocols at https://plos.org/protocols?utm_medium=editorial-email&utm_source=authorletters&utm_campaign=protocols..

We look forward to receiving your revised manuscript.

Kind regards,

Mehran Rahimlou, PhD

Academic Editor

PLOS ONE

Journal Requirements:

[This work was supported by General Program of the National Natural Science Foundation of China (82274455); Youth Program of the National Natural Science Foundation of China (82104805); Key Research Project of the Liaoning Provincial Department of Education (JYTZD2023196, JYTMS20231828); Joint Fund of the Liaoning Provincial Science and Technology Department (2023-MSLH-154, 2023-MSLH-147); Yin Yuanping National Famous Traditional Chinese Medicine Expert Inheritance Studio (202205); Reserve Project of Basic Scientific Research in Liaoning Colleges and Universities (2024-YJTCB-045).].

7. Please amend either the abstract on the online submission form (via Edit Submission) or the abstract in the manuscript so that they are identical.

Reviewers' comments:

Reviewer's Responses to Questions

**Comments to the Author**

1. Is the manuscript technically sound, and do the data support the conclusions?

Reviewer #1: Yes

Reviewer #2: Yes

Reviewer #3: Yes

2. Has the statistical analysis been performed appropriately and rigorously? 

Reviewer #1: Yes

Reviewer #2: No

Reviewer #3: Yes

3. Have the authors made all data underlying the findings in their manuscript fully available?

Reviewer #1: Yes

Reviewer #2: Yes

Reviewer #3: No

4. Is the manuscript presented in an intelligible fashion and written in standard English?

Reviewer #1: Yes

Reviewer #2: Yes

Reviewer #3: Yes

5. Review Comments to the Author

Reviewer #1: The manuscript Yue Luo et al. investigates the evaluation the therapeutic efficacy of a ketogenic diet in ameliorating sodium iodide-induced experimental autouimmune thyroiditis in murine models. Given that this study is able to conclude that the potential of ketogenetic diet to improve the autoimmune response, the results of this study will certainly be of interest for readers of Plos One.

The manuscript is generally well written, current issues are around clarification, provision of further explanations in some places requested.

Generally, there should be comments included on inflammatory values (and other analyses results) and whether they are in a low-medium-or high range.

Actually, in L3 one meta-analysis is mentioned - is there only one? IF not, why was this one selected?

Write the reference for this sentence In recent years, the incidence of AIT has been increasing year by year, and its etiology is related to genetic susceptibility, environment, diet, infection, and other factors (L5).

Figure 5.A. is difficult to see unless blow up. I feel it may be better to concentrate on the primary outcome measures.

Some other specific issues; add references in the method session.

Explain in below tables and figures, the statistical analysis used for the data.

Add more reference in meta analysis and systematic review, so the discussion would be deep interpretation and logical analysis of research results.

Reviewer #2: Summary

The summary should be shortened to present the main conclusions more concisely.

Introduction

The hypothesis is well stated, and the introduction section is appropriately structured.

Materials and Methods:

“NOD.H-2 h4 mice were selected as the ideal animal model for AIT. The mice were provided with free access to 0.05% sodium iodide water and were developed into an AIT animal model after 8 weeks.”

Was this distilled or ionized water? Was it confirmed whether the iodine intake was at the intended level?

“control group (ZC), normal diet (ND) and ketogenic diet group (KD)”

The specific diet compositions for these groups are not described. Additionally, the group names ZC and ND are too similar, which may lead to misinterpretation.

“A total of 60 NOD.H-2h4 mice at 8 weeks of age”…

What were their body weights?

Was a reference used when formulating the ketogenic diet?

“with no gender restriction”

This statement is inappropriate. Sex differences may influence susceptibility to metabolic conditions and thus affect the outcomes.

“a standard diet”

The nutritional composition of the diets should be presented, e.g., in a table similar to Table 1.

“If the data did not meet the criteria for normal distribution or equal variance, non-parametric tests were employed, with appropriate corrections for multiple comparisons”

The specific statistical tests used should be clearly stated.

Results, Discussion, and Conclusion:

These sections are well-structured.

Tables:

8. “The feed formulations”

Only one diet is provided. The diets of the “normal control group (ZC), normal diet (ND), and ketogenic diet group (KD)” are not defined. The nutrient compositions should be included.

What does “normol” mean?

Figures:

10. Although sodium iodide was administered for 8 weeks, the figure gives the impression that it was discontinued afterward. Is this correct? What was the rationale behind this implementation? The manuscript does not explain the reason for this interruption, and it should be clarified.

Reviewer #3: Abstract:

This study investigates the effects of the ketogenic diet on antioxidant capacity, immune response, and inflammation in a sodium iodide-induced autoimmune thyroiditis (AIT) mouse model. The aim of the study is to observe the anti-AIT effects of the ketogenic diet in iodine-induced NOD.H-2h4 mice and provide a theoretical basis for understanding the mechanism of the ketogenic diet in AIT treatment, particularly explained through the HMGB1/NLRP3/Caspase-1 signaling pathways.

Introduction:

The introduction provides an overview of the research topic, information on autoimmune thyroiditis (AIT), the mouse model used, and the potential benefits of the ketogenic diet in autoimmune diseases. The primary objective of the study is to examine the effects of the ketogenic diet on AIT in mice and understand its mechanisms, particularly through the HMGB1/NLRP3/Caspase-1 pathways. However, further detailed explanations of the autoimmune and pro-inflammatory markers used (e.g., TPOAb, TGAb, TNF-α, IL-6, IL-18) would help the reader better understand the significance of these markers and the study's focus.

Methods:

The methods section is well-structured, and the parameters and methods used are explained in detail. The investigation of autoimmune markers (TPOAb, TGAb) and pro-inflammatory cytokines (TNF-α, IL-6, IL-18) is critical for understanding immune response and inflammation. However, further clarification of the reasons for selecting these markers and how they are related to the effects of the ketogenic diet on AIT would improve the comprehensibility of the study. Additionally, providing more information about the experimental timeline and the dose-response relationship between sodium iodide and the ketogenic diet would enhance the reproducibility of the study.

Results:

The results are presented clearly, demonstrating that the ketogenic diet alleviated autoimmune thyroiditis induced by high iodine intake in mice. Specifically, the ketogenic diet was found to reduce inflammatory markers, decrease oxidative stress, and increase antioxidant levels. Moreover, the ketogenic diet reduced the expression of the NLRP3/ASC/Caspase-1 signaling pathway and corrected the Th17/Treg imbalance. These findings provide valuable insights into the potential effects of the ketogenic diet. However, reporting the statistical results and p-values not only under the figures but also within the text could further support the findings.

Discussion:

The discussion section links the study's findings to the existing literature, elucidating the potential mechanisms of the ketogenic diet's effects on AIT. The effects of the ketogenic diet on inflammatory cytokines, oxidative stress, and immune modulation are well explained. However, a deeper exploration of the modulation of the HMGB1/NLRP3/Caspase-1 pathways could provide more detailed information on how these pathways operate. Additionally, comparing the ketogenic diet with other autoimmune diseases or dietary interventions could help contextualize the findings in a broader scope. Discussing the potential clinical implications of these findings would also enhance the study's overall relevance.

Conclusion:

The conclusion effectively summarizes the key findings, emphasizing that the ketogenic diet alleviated autoimmune thyroiditis and restored immune balance. The study's limitations were acknowledged, and suggestions for future research were provided. Both the discussion and conclusion sections present the findings and their possible implications effectively.

Suggestions for Improvement:

1. Clarification of Markers: In the introduction, including a brief explanation of the autoimmune and pro-inflammatory markers (e.g., TPOAb, TGAb, TNF-α) would be beneficial for readers who may not be familiar with the subject.

2. Statistical Transparency: Including p-values or confidence intervals for the results in the results section would support the findings and provide readers with more insight into the reliability and significance of the data.

3. Mechanistic Review: The discussion section could be strengthened by a more in-depth examination of the effects of the ketogenic diet on the HMGB1/NLRP3/Caspase-1 signaling pathways. More information on how these pathways are modulated and their relationship with other autoimmune diseases or dietary interventions could be added.

4. Clinical Applications: Including a brief section on the clinical applications of the study, especially discussing the potential effects on human autoimmune diseases, would help the study appeal to a broader audience.

5. Limitations and Future Research: While the study's limitations were addressed, discussing potential confounding factors (e.g., differences in mouse strain, variations in diet composition) and recommending specific areas for future research would be important to expand the scope of the study.

6. PLOS authors have the option to publish the peer review history of their article (what does this mean?). If published, this will include your full peer review and any attached files.). If published, this will include your full peer review and any attached files.

.

Reviewer #1: **Yes:** Ratih Wirapuspita WisnuwardaniRatih Wirapuspita Wisnuwardani

Reviewer #2: No

Reviewer #3: No

While revising your submission, please upload your figure files to the Preflight Analysis and Conversion Engine (PACE) digital diagnostic tool, https://pacev2.apexcovantage.com/. PACE helps ensure that figures meet PLOS requirements. To use PACE, you must first register as a user. Registration is free. Then, login and navigate to the UPLOAD tab, where you will find detailed instructions on how to use the tool. If you encounter any issues or have any questions when using PACE, please email PLOS at . PACE helps ensure that figures meet PLOS requirements. To use PACE, you must first register as a user. Registration is free. Then, login and navigate to the UPLOAD tab, where you will find detailed instructions on how to use the tool. If you encounter any issues or have any questions when using PACE, please email PLOS at figures@plos.org. Please note that Supporting Information files do not need this step.. Please note that Supporting Information files do not need this step.

---

## [Author Response · Author response to Decision Letter 1]

14 Jul 2025

Journal Requirements:

Response 1: It has been modified as required.

Response 2: Mice were euthanized through deep anesthesia induced by an intraperitoneal injection of sodium pentobarbital. Subsequently, blood was collected from the orbital vein, and the thyroid gland was isolated via cervical dissection. Animal welfare was given top priority throughout the study. This was achieved by implementing environmental enrichment measures, determining statistically optimized sample sizes, and strictly adhering to the 3Rs principles (Replacement, Reduction, and Refinement) as well as the ARRIVE (Animal Research: Reporting of In Vivo Experiments) ethical guidelines

In the "Sample Collection" section of the "Materials and Methods," the following content has been added: All experimental procedures adhered to institutional ethical guidelines and the 3Rs principles. Mice were deeply anesthetized via intraperitoneal sodium pentobarbital (65 mg/kg; 10 mL/kg). Blood was collected from the orbital vein using sterilized capillaries post-mydriasis (0.5% phenylephrine). Serum was separated by centrifugation (2000 × g, 15 min) after coagulation and stored in 1.5 mL tubes at -80° C. Thyroid glands were isolated through midline cervical dissection: laryngeal-tracheal complexes were exposed following blunt submandibular gland retraction, with meticulous preservation of parathyroids. Tissues were either fixed in 4% paraformaldehyde or snap-frozen in liquid nitrogen; RT-qPCR samples were preserved with 1 mL DNA extraction reagent prior to -80° C storage. Western blot specimens were directly immersed in liquid nitrogen. Environmental enrichment (nesting materials, social housing) and power analysis-driven sample minimization ensured welfare optimization. The study received ethical approval and complied with ARRIVE guidelines.

[This work was supported by General Program of the National Natural Science Foundation of China (82274455); Youth Program of the National Natural Science Foundation of China (82104805); Key Research Project of the Liaoning Provincial Department of Education (JYTZD2023196, JYTMS20231828); Joint Fund of the Liaoning Provincial Science and Technology Department (2023-MSLH-154, 2023-MSLH-147); Yin Yuanping National Famous Traditional Chinese Medicine Expert Inheritance Studio (202205); Reserve Project of Basic Scientific Research in Liaoning Colleges and Universities (2024-YJTCB-045).].

Response 3-4: We have revised the "Funding Information" section to ensure consistency and accuracy regarding the grant numbers and sources of support for our study. The funding for this work was provided by the following sources: General Program of the National Natural Science Foundation of China (82274455); Youth Program of the National Natural Science Foundation of China (82104805); Key Research Project of the Liaoning Provincial Department of Education (JYTZD2023196, JYTMS20231828); Joint Fund of the Liaoning Provincial Science and Technology Department (2023-MSLH-154, 2023-MSLH-147); Yin Yuanping National Famous Traditional Chinese Medicine Expert Inheritance Studio (202205); Reserve Project of Basic Scientific Research in Liaoning Colleges and Universities (2024-YJTCB-045). We confirm that there was no additional external funding received for this study.

We have confirmed the details of the Financial Disclosure. We have attached additional information regarding this in both the cover letter and the "Funding Information & Financial Disclosure" document.

Response 5-6: We have made the uncropped gel images and all available data for the article publicly accessible in Data Source 1.

7. Please amend either the abstract on the online submission form (via Edit Submission) or the abstract in the manuscript so that they are identical.

Response 7: An updated version of the abstract has now been submitted, and it is consistent with the latest manuscript.

Reviewer #1/ Question 1: Generally, there should be comments included on inflammatory values (and other analyses results) and whether they are in a low-medium-or high range.

Reviewer #1/Response 1: In the section "2. Effects of ketogenic diet on serum cytokines and oxidative stress markers," the following information has been supplemented: Elevated serum concentrations of MCP-1, TNF-α, IL-1β, IL-6, IL-18, IFN-γ, and MDA indicate acute inflammatory responses, whereas reduced levels of SOD and T-AOC suggest a correlation between impaired antioxidant capacity and exacerbated inflammatory processes.

Reviewer #1/ Question 2: Actually, in L3 one meta-analysis is mentioned - is there only one? IF not, why was this one selected?

Reviewer #1/Response 2: This study further supplements meta-analyses such as those with PMIDs: 39641795, 39356749, and 39135006. It systematically evaluated and conducted a meta-analysis on the global prevalence trends of autoimmune thyroiditis (AIT) over time, providing crucial evidence for the formulation of global public health strategies. With its rigorous methodology and broader coverage of studies, this research was selected for citation.”

Reviewer #1/ Question 3: Write the reference for this sentence In recent years, the incidence of AIT has been increasing year by year, and its etiology is related to genetic susceptibility, environment, diet, infection, and other factors (L5).

Reviewer #1/Response 3: An apology is made for the omission of the reference, which has now been rectified and supplemented as follows: Kravchenko V, Zakharchenko T. Thyroid hormones and minerals in immunocorrection of disorders in autoimmune thyroid diseases. Front Endocrinol (Lausanne). 2023;14:1225494. Published on August 30, 2023. doi:10.3389/fendo.2023.1225494.

Reviewer #1/ Question 4: Figure 5.A. is difficult to see unless blow up. I feel it may be better to concentrate on the primary outcome measures.

Reviewer #1/Response 4: The clear version of the original Figure 5.A has been re-updated. Since Figure 5.A might have included secondary outcome measures, the original Figure 5.A has been placed in the appendix or supplementary materials (Supplementary Figure 1). Additionally, the textual sections of the thesis have been examined to ensure that the primary outcome measures are prominently highlighted.

Reviewer #1/ Question 5: Some other specific issues; add references in the method session.

Reviewer #1/Response 5: In the "Model Establishment and Intervention" section, a reference was supplemented for the model assessment (The mice were granted free access to 0.05% sodium iodide water and were subsequently developed into an AIT animal model after 8 weeks): Braley-Mullen H, Sharp GC, Medling B, Tang H. Spontaneous autoimmune thyroiditis in NOD.H-2h4 mice. J Autoimmun. 1999;12(3):157-165. doi:10.1006/jaut.1999.0272. In the "Sample Collection" section, a reference was supplemented regarding the dosage of the anesthetic agent: George FR, Jackson SJ, Collins AC. Prostaglandin synthetase inhibitors antagonize hypothermia induced by sedative hypnotics. Psychopharmacology (Berl). 1981;74(3):241-244. doi:10.1007/BF00427102. In the "ELISA Assay" section, a reference was supplemented for the ELISA experimental protocol: Li F, Zhou Z, Wang L, et al. A study of programmed death-1/programmed death ligand and iodine-induced autoimmune thyroiditis in NOD.H-2h4 mice. Environ Toxicol. 2023;38(11):2574-2584. doi:10.1002/tox.23893.

Reviewer #1/ Question 6: Explain in below tables and figures, the statistical analysis used for the data.

Reviewer #1/Response 6: Provide a detailed description of the statistical method, update the "Statistical Analysis" section and explain the statistical analysis used for the data in the tables and figures below. For example, P-value from one-way ANOVA test

The updated content is as follows: All experimental data in this study were analyzed using Excel, SPSS 22.0, and GraphPad Prism 8 statistical software. Initially, the normality of data distribution was assessed via the Shapiro-Wilk test. For normally distributed data, independent samples t-tests were employed for intergroup comparisons; for non-normally distributed data, nonparametric Mann-Whitney U tests (rank-sum tests) were utilized. In comparisons involving multiple groups, one-way analysis of variance (ANOVA) was applied when data conformed to normal distribution with homogeneous variances, while the nonparametric Kruskal-Wallis H test (rank-sum test) was used when variances were heterogeneous.

Statistical significance was defined as P < 0.05 (two-tailed tests). All analytical processes were based on raw data, with graphical and numerical outputs generated through SPSS 22.0. Final statistical conclusions were drawn based on significance levels (P < 0.05) to determine whether differences between experimental and control groups were statistically meaningful. Quantitative results are presented as mean ± standard deviation (Mean ± SD), with intergroup comparisons visually supplemented by figures/tables as required.

Reviewer #1/ Question 7: Add more reference in meta analysis and systematic review, so the discussion would be deep interpretation and logical analysis of research results.

Reviewer #1/Response 7: Consistent with Reviewer # 1/Response 2. This study further supplements meta-analyses such as those with PMIDs: 39641795, 39356749, and 39135006. It systematically evaluated and conducted a meta-analysis on the global prevalence trends of autoimmune thyroiditis (AIT) over time, providing crucial evidence for the formulation of global public health strategies. With its rigorous methodology and broader coverage of studies, this research was selected for citation.

Reviewer #2:

Reviewer #2/ Question 1: Summary： The summary should be shortened to present the main conclusions more concisely.

Reviewer #2/Response 1: Further abbreviate the abstract and ensure the presentation of the main content and conclusions, as follows: Background: Autoimmune thyroiditis (AIT), caused by immune-mediated thyroid dysfunction, lacks effective dietary interventions. This study investigates the therapeutic potential of a ketogenic diet (KD) in a murine model of iodine-induced AIT. Methods: Sixty 8-week-old NOD.H-2h4 mice were randomized into three groups: wild-type mice on a normal diet (WT+ND), AIT model mice on a normal diet (AIT+ND), and AIT model mice on a KD (AIT+KD). AIT was induced in model groups via 0.05% sodium iodide administration for 8 weeks, followed by 4 weeks of KD intervention in the AIT+KD group. Outcomes included histopathological thyroid changes, serum levels of TgAb/TPOAb, proinflammatory cytokines (MCP-1, TNF-α, IL-1β, IL-6, IL-18, IFN-γ), oxidative stress markers (MDA, SOD, T-AOC), gene expression of NLRP3 inflammasome components (NLRP3, ASC, Caspase-1), Th17/Treg-related factors (RORγt, IL-17, FoxP3, IL-10), and immune mediators (TLR2/4, NF-κB, HMGB1). Results: Compared to WT+ND, the AIT+ND group showed elevated TgAb/TPOAb, thyroid lymphocytic infiltration, increased proinflammatory cytokines, and oxidative stress. Upregulated NLRP3 inflammasome activity, Th17 polarization, and HMGB1/TLR2/4/NF-κB signaling were observed, alongside a higher Th17/Treg ratio. KD intervention reversed these alterations, suppressing inflammation, oxidative stress, and pathogenic immune pathways. Conclusion: KD ameliorates iodine-induced AIT in mice by modulating HMGB1/NLRP3-mediated inflammation, restoring immune balance, and reducing thyroid autoimmunity. These findings support KD as a potential adjuvant therapy for AIT, warranting clinical evaluation.

Reviewer #2/ Question 2: Was this distilled or ionized water? Was it confirmed whether the iodine intake was at the intended level?

Reviewer #2/ Response 2: This study used double distilled water and 0.05% sodium iodide (NaI) for 8 weeks. Similar situations in the text have been modified. As can be seen from previous research literature: Yang X, Gao T, Shi R, et al. Effect of iodine excess on Th1, Th2, Th17, and Treg cell subpopulations in the thyroid of NOD.H-2h4 mice. Biol Trace Elem Res. 2014; 159 (1-3): 288-296. doi: 10.1007/s12011-014-9958-y, Zhao Z, Liu Z, Song N, et al. Based on SIRT1/NF - κ B/NLRP3 Signal Pathway to Explore the Effect of Yiqi Huatan Huoxue Recipe on Inflammation Injury in AIT Mice by Pyroptosis. Comb Chem High Throughput Screen. Published online August 12, 2024. doi: 10.2174/01138620732948240801113424 (reference added).

Reviewer #2/ Question 3: “control group (ZC), normal diet (ND) and ketogenic diet group (KD)”. The specific diet compositions for these groups are not described. Additionally, the group names ZC and ND are

---

## [Decision Letter · Decision Letter 1]

4 Sep 2025

PONE-D-25-04384R1The Ketogenic Diet Alleviates Autoimmune Thyroiditis Caused by Th17/Treg Imbalance by Inhibiting the HMGB1/NLRP3 Signaling PathwayPLOS ONE

Dear Dr. Yang,

Thank you for submitting your manuscript to PLOS ONE. After careful consideration, we feel that it has merit but does not fully meet PLOS ONE’s publication criteria as it currently stands. Therefore, we invite you to submit a revised version of the manuscript that addresses the points raised during the review process.

If applicable, we recommend that you deposit your laboratory protocols in protocols.io to enhance the reproducibility of your results. Protocols.io assigns your protocol its own identifier (DOI) so that it can be cited independently in the future. For instructions see: https://journals.plos.org/plosone/s/submission-guidelines#loc-laboratory-protocols. Additionally, PLOS ONE offers an option for publishing peer-reviewed Lab Protocol articles, which describe protocols hosted on protocols.io. Read more information on sharing protocols at . Additionally, PLOS ONE offers an option for publishing peer-reviewed Lab Protocol articles, which describe protocols hosted on protocols.io. Read more information on sharing protocols at https://plos.org/protocols?utm_medium=editorial-email&utm_source=authorletters&utm_campaign=protocols..

We look forward to receiving your revised manuscript.

Kind regards,

Mehran Rahimlou, PhD

Academic Editor

PLOS ONE

**Journal Requirements:**

Reviewers' comments:

Reviewer's Responses to Questions

**Comments to the Author**

1. If the authors have adequately addressed your comments raised in a previous round of review and you feel that this manuscript is now acceptable for publication, you may indicate that here to bypass the “Comments to the Author” section, enter your conflict of interest statement in the “Confidential to Editor” section, and submit your "Accept" recommendation.

Reviewer #1: (No Response)

Reviewer #2: All comments have been addressed

Reviewer #3: All comments have been addressed

2. Is the manuscript technically sound, and do the data support the conclusions?

Reviewer #1: Partly

Reviewer #2: Yes

Reviewer #3: Yes

3. Has the statistical analysis been performed appropriately and rigorously? 

Reviewer #1: Yes

Reviewer #2: Yes

Reviewer #3: Yes

4. Have the authors made all data underlying the findings in their manuscript fully available?

Reviewer #1: Yes

Reviewer #2: Yes

Reviewer #3: Yes

5. Is the manuscript presented in an intelligible fashion and written in standard English?

Reviewer #1: Yes

Reviewer #2: Yes

Reviewer #3: Yes

6. Review Comments to the Author

**Reviewer #1:** Yue Luo et al. wrote an article aiming to investigate the therapeutic potential of a ketogenic diet in ameliorating iodine-induced autoimmune thyroiditis in mice, focusing on its effects on inflammation, oxidative stress, immune pathways, and thyroid autoimmunity. I believe this article may be of interest to the PLOS ONE, but several aspects still require improvement to strengthen its scientific contribution. Yue Luo et al. wrote an article aiming to investigate the therapeutic potential of a ketogenic diet in ameliorating iodine-induced autoimmune thyroiditis in mice, focusing on its effects on inflammation, oxidative stress, immune pathways, and thyroid autoimmunity. I believe this article may be of interest to the PLOS ONE, but several aspects still require improvement to strengthen its scientific contribution.

First, the explanation of the research aims needs to be clarified. Furthermore, the description of ketogenic diet requires more detail. In particular, the article should explain how ketogenetic diet, inflammation, oxidative stress, immune pathways, and thyroid autoimmunity were treated during the experience. This is important to ensure the validity and reproducibility of the variables estimates.

Second, the method used has the weakness. Please explain it in the discussion section, as ELISA Kit has the limitation. In addition, the limitations of using ELISA Kit to assess IL should be acknowledged in more depth. Specifically, the potential measurement bias associated with other measurements, should be discussed both in the discussion and limitations sections.

ABSTRACT

Please consider revising the RESULTS section of the abstract to keep it concise, and clear

Tables 1 and 2 have been revised for better clarity and are now left-aligned, formatted according to the journal guidelines

**Reviewer #2:** "There is an error in Table 1, the normal diet is unclear. The group names in the figures remain the same as before." "There is an error in Table 1, the normal diet is unclear. The group names in the figures remain the same as before."

**Reviewer #3:** (No Response)(No Response)

7. PLOS authors have the option to publish the peer review history of their article (what does this mean?). If published, this will include your full peer review and any attached files.). If published, this will include your full peer review and any attached files.

.

Reviewer #1: **Yes:** Ratih Wirapuspita WisnuwardaniRatih Wirapuspita Wisnuwardani

Reviewer #2: No

Reviewer #3: No

While revising your submission, please upload your figure files to the Preflight Analysis and Conversion Engine (PACE) digital diagnostic tool, https://pacev2.apexcovantage.com/. PACE helps ensure that figures meet PLOS requirements. To use PACE, you must first register as a user. Registration is free. Then, login and navigate to the UPLOAD tab, where you will find detailed instructions on how to use the tool. If you encounter any issues or have any questions when using PACE, please email PLOS at . PACE helps ensure that figures meet PLOS requirements. To use PACE, you must first register as a user. Registration is free. Then, login and navigate to the UPLOAD tab, where you will find detailed instructions on how to use the tool. If you encounter any issues or have any questions when using PACE, please email PLOS at figures@plos.org. Please note that Supporting Information files do not need this step.. Please note that Supporting Information files do not need this step.

---

## [Author Response · Author response to Decision Letter 2]

17 Oct 2025

Reviewer #1:

Comment 1.1: First, the explanation of the research aims needs to be clarified. Furthermore, the description of ketogenic diet requires more detail. In particular, the article should explain how ketogenetic diet, inflammation, oxidative stress, immune pathways, and thyroid autoimmunity were treated during the experience. This is important to ensure the validity and reproducibility of the variables estimates.

Response 1.1: We would like to express our sincere gratitude to the reviewer for your insightful comments and valuable feedback. The explanation of the research aims has been added to the text, is in the Manuscript file Page 6, line 125-129. The explanation of the validity and reproducibility of the variables estimates has been added to the text, is in the Manuscript file Page 15-16, line 404-408.

Text: The research aims: the purpose of this study was to observe the anti-AIT effect of KD on iodine-induced NOD.H-2h4 mice and for the first time discussed its mechanism of action based on the HMGB1/NLRP3/Caspase-1 signaling pathway, providing a theoretical basis for the mechanism study of KD in the treatment of AIT.

The explanation of the validity and reproducibility of the variables estimates: In the present study, the efficacy of the KD intervention was validated by defining the KD formulation in detail. Outcome variables were assessed using multi-dimensional indices, including both serum and tissue samples, as well as protein and mRNA levels, thereby ensuring the reliability of the results.

Comment 1.2: Second, the method used has the weakness. Please explain it in the discussion section, as ELISA Kit has the limitation. In addition, the limitations of using ELISA Kit to assess IL should be acknowledged in more depth. Specifically, the potential measurement bias associated with other measurements, should be discussed both in the discussion and limitations sections.

Response 1.2: The explanation of method used has been added in the discussion section in the Manuscript file Page16-17, line 430-436 and Study Limitations in Page19-20, line516-519.

The discussion section: The enzyme-linked immunosorbent assay (ELISA) is widely used for interleukin detection in biomedical research due to its high throughput, operational convenience, and relatively low cost. However, this method has inherent limitations that may introduce measurement bias, such as ELISA quantifies the total protein level of ILs but cannot differentiate between biologically active cytokines and their inactive precursors, degradation products, or cytokine-antibody complexes, furthermore especially when compared with alternative techniques such as FCM, qRT-PCR, and Western blotting. Thus, the aforementioned techniques were employed to address these limitations.

Study Limitations: specificity-related bias in ELISA may mislead the interpretation of inflammatory response intensity in autoimmune thyroiditis (AIT) models, and future studies can be extended to multiple mouse strains to capture multiple response patterns that match the diversity of human AIT.

Comment 1.3:ABSTRACT

Please consider revising the RESULTS section of the abstract to keep it concise, and clear

Response 1.3: The RESULTS section of the abstract is in the Manuscript file Page 2, line 46-53.

Text: Results: Compared to the WT+ND group, the AIT+ND group exhibited increased TgAb/TPOAb levels, enhanced thyroid lymphocytic infiltration, elevated proinflammatory cytokines, and aggravated oxidative stress. Concurrently, this group showed upregulated NLRP3 inflammasome activity, promoted Th17 polarization, activated HMGB1/TLR2/4/NF-κB signaling, and a higher Th17/Treg ratio. Ketogenic diet (KD) intervention reversed all these alterations, effectively suppressing inflammation, oxidative stress, and pathogenic immune pathways.

Comment 2: There is an error in Table 1, the normal diet is unclear. The group names in the figures remain the same as before.

Response 2: Table 1 has been modified in the Manuscript file Page 29, line 749-750.

---

## [Decision Letter · Decision Letter 2]

2 Jan 2026

PONE-D-25-04384R2The Ketogenic Diet Alleviates Autoimmune Thyroiditis Caused by Th17/Treg Imbalance by Inhibiting the HMGB1/NLRP3 Signaling PathwayPLOS One

Dear Dr. Yang,

Thank you for submitting your revised manuscript to PLOS ONE. After careful consideration, we feel that it has merit but does not fully meet PLOS ONE’s publication criteria as it currently stands. Therefore, we invite you to submit a revised version of the manuscript that addresses the points raised by the reviewer.

If applicable, we recommend that you deposit your laboratory protocols in protocols.io to enhance the reproducibility of your results. Protocols.io assigns your protocol its own identifier (DOI) so that it can be cited independently in the future. For instructions see: https://journals.plos.org/plosone/s/submission-guidelines#loc-laboratory-protocols. Additionally, PLOS ONE offers an option for publishing peer-reviewed Lab Protocol articles, which describe protocols hosted on protocols.io. Read more information on sharing protocols at . Additionally, PLOS ONE offers an option for publishing peer-reviewed Lab Protocol articles, which describe protocols hosted on protocols.io. Read more information on sharing protocols at https://plos.org/protocols?utm_medium=editorial-email&utm_source=authorletters&utm_campaign=protocols..

We look forward to receiving your revised manuscript.

Kind regards,

Kota V Ramana, Ph.D.

Academic Editor

PLOS One

Journal Requirements:

Reviewers' comments:

Reviewer's Responses to Questions

**Comments to the Author**

1. If the authors have adequately addressed your comments raised in a previous round of review and you feel that this manuscript is now acceptable for publication, you may indicate that here to bypass the “Comments to the Author” section, enter your conflict of interest statement in the “Confidential to Editor” section, and submit your "Accept" recommendation.

Reviewer #1: All comments have been addressed

Reviewer #2: All comments have been addressed

Reviewer #3: (No Response)

2. Is the manuscript technically sound, and do the data support the conclusions?

Reviewer #1: Yes

Reviewer #2: Yes

Reviewer #3: Yes

3. Has the statistical analysis been performed appropriately and rigorously? 

Reviewer #1: Yes

Reviewer #2: Yes

Reviewer #3: Yes

4. Have the authors made all data underlying the findings in their manuscript fully available?

Reviewer #1: Yes

Reviewer #2: Yes

Reviewer #3: Yes

5. Is the manuscript presented in an intelligible fashion and written in standard English?

Reviewer #1: Yes

Reviewer #2: Yes

Reviewer #3: Yes

6. Review Comments to the Author

Reviewer #1: This paper, authored by Xiao Yang et al., investigates the therapeutic potential of a ketogenic diet (KD) in a murine model of iodine-induced Autoimmune Thyroiditis (AIT). This topic is highly relevant to the scope of PLOS ONE due to its translational focus on diet modification for managing autoimmune disorders.

Abstract

Mention the data analysis used. What was the percentage increase in TgAb/TPOAb levels, enhanced thyroid lymphocytic infiltration, elevated proinflammatory cytokines, and aggravated oxidative stress? What was the p-value for the difference?

Methods

How many mice were used?

Discussion

Add a section at the end of the session for the strengths and weaknesses of the research.

Reviewer #2: (No Response)

Reviewer #3: Based on the reviewer's comments, I see that the necessary corrections have been made appropriately. I did not encounter any errors in spelling, punctuation, or other grammatical rules during my evaluation

7. PLOS authors have the option to publish the peer review history of their article (what does this mean?). If published, this will include your full peer review and any attached files.). If published, this will include your full peer review and any attached files.

.

Reviewer #1: **Yes:** Ratih Wirapuspita WisnuwardaniRatih Wirapuspita Wisnuwardani

Reviewer #2: No

Reviewer #3: No

---

## [Author Response · Author response to Decision Letter 3]

7 Jan 2026

REVIEWER #1.1: Mention the data analysis used. What was the percentage increase in TgAb/TPOAb levels, enhanced thyroid lymphocytic infiltration, elevated proinflammatory cytokines, and aggravated oxidative stress? What was the p-value for the difference?

Reply 1: The tables below summarize the changes in TgAb levels, TPOAb levels, thyroid lymphocytic infiltration index, pro-inflammatory cytokines levels, and oxidative stress levels, respectively. (Page 12, line 311 and Page 13, line 327 and Page 13, line 335 )

REVIEWER #1.2: How many mice were used?

Reply 1.2: Sixty 8-week-old SPF NOD.H-2h4 mice were stratified by body weight and then completely randomized into three experimental groups (n = 20 per group). This protocol satisfies the 3R principle while ensuring adequate statistical power.

REVIEWER #1.3: Add a section at the end of the session for the strengths and weaknesses of the research.

Reply 1.3: Using the international iodine-excess NOD.H-2h4 standard, we established a rigorous AIT model and provide the first integrated evidence that a ketogenic diet simultaneously suppresses the HMGB1–TLR/NF-κB–NLRP3 inflammasome axis and rebalances the Th17/Treg compartment, markedly attenuating thyroid damage and circulating autoantibody levels. These findings offer a novel dietary strategy for AIT management. Nevertheless, the present work is confined to in vivo proof-of-concept, and the precise molecular targets and regulatory hierarchies remain to be definitively elucidated in future studies. ((Page 20, line 536 )

---

## [Editor Report · Decision Letter 3]

8 Jan 2026

The Ketogenic Diet Alleviates Autoimmune Thyroiditis Caused by Th17/Treg Imbalance by Inhibiting the HMGB1/NLRP3 Signaling Pathway

PONE-D-25-04384R3

Dear Dr. Yang,

We’re pleased to inform you that your manuscript has been judged scientifically suitable for publication and will be formally accepted for publication once it meets all outstanding technical requirements.

An invoice will be generated when your article is formally accepted. Please note, if your institution has a publishing partnership with PLOS and your article meets the relevant criteria, all or part of your publication costs will be covered. Please make sure your user information is up-to-date by logging into Editorial Manager at Editorial Manager® and clicking the ‘Update My Information' link at the top of the page. For questions related to billing, please contact  and clicking the ‘Update My Information' link at the top of the page. For questions related to billing, please contact billing support..

Kind regards,

Kota V Ramana, Ph.D.

Academic Editor

PLOS One
---

## [Editor Report · Acceptance letter]

PONE-D-25-04384R3

PLOS One

Dear Dr. Yang,

I'm pleased to inform you that your manuscript has been deemed suitable for publication in PLOS One. Congratulations! Your manuscript is now being handed over to our production team.

Kind regards,

on behalf of

Dr. Kota V Ramana

Academic Editor

PLOS One